# Work from Home during the COVID-19 Pandemic—The Impact on Employees’ Self-Assessed Job Performance

**DOI:** 10.3390/ijerph191710935

**Published:** 2022-09-01

**Authors:** Claudiu Vasile Kifor, Roxana Florența Săvescu, Raluca Dănuț

**Affiliations:** Research Center for Sustainable Products and Processes, Lucian Blaga University of Sibiu, 550143 Sibiu, Romania

**Keywords:** work from home, social performance, technical performance, COVID-19 pandemic, quantile regression analysis

## Abstract

This study investigates the impact of remote workplace factors on employees’ social and technical self-assessed performance during the COVID-19 pandemic. The impact of the variables belonging to the employee’s profile, organizational environment, and work-life balance categories on social and technical performance were analyzed, based on a survey of 801 Romanian employees, using ordinary least squares and quantile regression techniques. While the first method provided summary point estimates that calculated the average effect of the explanatory variables for the “average employee”, the second approach allowed us to focus on the effects explanatory variables have on the entire conditional distribution of the response variables, taking into account that this effect can be different for employees with different levels of performance. Job autonomy, engagement, communication skills, trust in co-workers, occupational self-efficacy, and family-work conflict, significantly influence both social and technical performance. PhD education and trust in management significantly influence social performance, while motivation, stress, the share of time spent in remote work, organizational commitment, children in the household, and household size, influence only technical performance.

## 1. Introduction

The COVID-19 pandemic created an unprecedented context for the expansion of remote work. Companies and employees alike had to adjust quickly to the new form of work organization, with employees struggling to juggle sometimes competing demands from their families, while trying to remain focused on job tasks and communicate efficiently with their colleagues, all against the backdrop of growing anxieties triggered by the pandemic [1]. Companies aimed to make the transition to remote work as smooth as possible, fearing drops in work productivity and interruptions in collaboration flows [2].

The new setup forced a digital revolution within companies; home offices had to be equipped with the devices needed for employees to carry out their work duties, and IT infrastructures had to be updated to include new security protocols, all in a time crunch. The living space was divided between work and leisure, and boundaries between the two blurred. Conventional work meetings were quickly replaced with video calls, and informal conversations with colleagues moved to dedicated chat rooms.

Working remotely was not common in Romania before the pandemic. The legislation on teleworking was first approved in 2018, when Romania had one of the lowest percentages of remote work in the EU (European Union), 0.4% [3].

However, by the fall of 2020, Romanian media was filled with quotes from CEOs and managers praising the new work organization and claiming that it not only protected employees from getting sick, but also saved companies money by optimizing office spaces and cutting down maintenance costs, with no fall in productivity [4]. 

As work from home (WFH) continued well after the COVID-19 pandemic, questions emerged: Are employees able to maintain a healthy balance between work and personal life? Can they stay productive when working from home, among distractions from family members, or while balancing household chores? Have they experienced feelings of isolation or felt dissatisfied with their work during prolonged periods of working remotely? Answering these questions is paramount to ensuring that this work organization is feasible in the long term, as many companies seem to suggest that “work from anywhere” is the future of employment [5].

This paper aims to investigate the impact of remote work on employees’ self-assessed performance when WFH during the COVID-19 pandemic. The article is organized as follows. Section 2 presents the literature review results concerning factors influencing adjustment to remote work, and formulates the research questions. Section 3 introduces the study design. Job characteristics specific to employee’s profile, organizational environment, and work-life balance were measured in a quantitative survey with 801 employees from Romania, and the results were analyzed using ordinary least squares (OLS) and quantile regression (QR) models in Section 4. A discussion of the findings concludes the paper in Section 5 and Section 6. The study results are helpful for researchers and human resources specialists in both academic and cross-business fields.

## 2. Literature Review ad Research Questions

In the early 1990s, Toffler [6] and Handy [7] anticipated that WFH would become a usual way of working, with benefits for the organization regarding job performance and employee satisfaction. Since the onset of the COVID-19 pandemic, the idea of WFH has spread rapidly, as it was, in some cases, the only way to maintain organizational permanence [8].

When the organization’s performance and productivity are under discussion in the WFH system, opinions are often contradictory. A study on WFH during the COVID-19 pandemic showed that WFH is associated with low employee productivity, and most workers experienced declines in productivity, probably due to their inadequate preparation for WFH under the sudden shock of the pandemic [9] (p. 21). Poor communication and lack of a well-developed setup for WFH were also mentioned among the factors that negatively influence employee productivity. On the opposite side, more recent studies showed that working from home brings more productivity among employees who practice it. Among the positive aspects, “the possibility to work from the comfort of one’s own home” and “time and money saved on commuting” are indicated [10,11,12] (pp. 6–7).

Abramis defined job performance as a “worker’s effective execution of tasks or job and useful contribution to the social work environment”. Abramis also introduced different dimensions of job performance, such as technical and social performance. The first one refers to “a worker’s handling of demands, making correct decisions, and performing without mistakes”, while social performance refers to “a worker’s ability to get along with others at work, make compromises, and avoid fighting or arguing” [13] (p. 549). 

In the traditional Work From Office (WFO) environments, maximum performance is believed to occur when the employee’s profile (gender, age, education knowledge, competencies, values, and interests) is consistent with the organizational environment and the needs of the job demands [14]. In remote working systems, it is important to understand also the work-life balance, work-life conflicts, and psychological well-being of teleworkers and the way these factors impact job performance [15].

### 2.1. Employee’s Profile and Job Performance 

Literature provides significant differences when analyzing the relationship between gender and job performance. Some studies showed that men seem to enjoy WFH, appreciate daily activities, experience less stress, and feel more able to overcome difficulties, while women reported lower satisfaction in family life and also lower work productivity [10,16,17], [11] (p. 27). On the other hand, Gajendran [18] found no evidence, while Allen et al. [19] and Sandoval-Reyes, J. et al. [20] show little evidence about the role played by gender in job performance, work-family conflict, or stress.

The *educational level* (secondary, undergraduate, graduate) has no significant effect on job performance via specifications, perceptions, and attitudes [10]. At the same time, the employees practicing WFH are more likely to be university graduates working in high-skilled employment [11]. 

At the same time, the employees practicing WFH are more likely to be university graduates working in high-skilled employment [21]. 

### 2.2. Organizational Environment and Job Performance 

*Work engagement* is defined as a condition characterized by energy, strong involvement, and complete focus [22,23]. A recent study showed that employees feel more engaged with their work because working from home affords them autonomy, safety and convenience during the COVID-19 pandemic, and that this work engagement leads to happiness [24] (p. 9). 

*Computer-mediated (C-M) communication skills* represent an essential element that ensures the good functioning of WFH activity. C-M communication knowledge is defined as ”the cognitive comprehension of content and procedural processes involved in concluding appropriate and effective interaction in the C-M context” [25] (p. 641). Other skills that people who work from home must have are: efficiency, proactivity, availability to respond to emails, and compliance with deadlines [11] (p. 24).

*Occupational self-efficacity or effectiveness* can be described as an individual’s belief in his or her own capacity to deal with complex assignments or challenges [26]. Individuals with higher effectiveness seek more challenging tasks, leading to higher performance [27] (p. 240).

The expectancy theory has a broader perspective: it examines the function of *motivation* in the work environment, instead of concentrating on personal wants, objectives, or social comparisons. The idea essentially holds that the way people perceive their role in the organization, influences their work performance [28,29,30] (p. 214). 

Organizational culture is defined as “a cognitive framework consisting of attitudes, values, behavioral norms, and expectations” [28] (p. 498). Previous studies found that conflict, solidarity, creativity, and goal clarity, all parts of the organizational culture, are significant predictors of productivity [28,31] (p. 180, p. 173). Workplace performance may increase when the goals set by the organization and managers are specific and accompanied by feedback. It should be noted that this feedback must be composed of motivational and informational elements [32]. Ed Locke and Gary Latham developed the theory of goal setting and task performance, which showed that a goal acts as a motivator by making employees evaluate their current performance concerning what is necessary to reach the goal [33]. 

*Organizational commitment* is the degree to which an employee identifies with the organization and wants to continue actively participating in it [34], being directly linked with turnover; employees who are firmly committed are those who are least likely to leave the organization [35].

Previous studies have emphasized the positive effects of organizational commitment on job performance [36,37,38]. The more committed employees tend to perform well and have a lower tendency to leave their jobs [39]. On the other hand, Taboroši S., et al. [40] are warning that in the WFH environment, the turnover is higher for the employees with low organizational engagement. Male teleworkers especially have a shallow level of organizational loyalty and the tendency to leave the organization if a better position is offered in another organization.

Al-Omari and Okashe showed that productivity could be influenced by *situational constraints*—multiple variables such as noise, office furniture, ventilation, temperature and light [41] (p. 15548). 

*Trust in co-workers* can be defined as “the willingness of a person to be vulnerable to the actions of fellow workers whose behavior and actions that persons cannot control” [42]. Co-workers refers to members of an organization who hold relatively equal power of authority and with whom an employee interacts during the workday. In other studies, it was observed that trust in co-workers is positively related to performance [42,43].

*Trust in management* is related to employees’ ability to focus attention on value-producing activities, and is subsequently related to a multifaceted treatment of performance [44].

Cook and Wall showed that the development of interpersonal trust in the workplace refers primarily to the trust the employee offers to others, correctly assessing their good intentions [45]. Very strictly monitored employees reported anxiety, burnout, and dissatisfaction [46]. In such cases, trust of any kind tends to be low [47]. Previous organizational studies have shown that trust in management generates a direct positive effect on several measures of job performance, while others have indicated no relationship [44,48].

### 2.3. Work-Life Balance and Job Performance 

Work-life balance is defined as a combination of factors such as family-work and work-family conflicts, psychological well-being, professional isolation, job autonomy, job satisfaction, and stress. Previous studies showed a significant correlation of work-life balance with employee attitudes and engagement, which positively correlates with job performance [49,50]. 

The challenge for employees working from home occurs when they have to play two different roles in the same space. The first involves job responsibilities, with deadlines and challenges from employers, and the second refers to family life and household needs [21]. When the employees fail to distinguish between the two roles, or when the requests coming from work and family are not compatible, the risk of work-home conflicts is inevitable [51,52]. *Work-family conflict* is a source of stress and has been linked to harmful effects, including physical and mental illness [51,53] (p. 206). Such employees, in turn, are less embedded in their jobs and display poor performance in the service delivery process [54].

In order to avoid these problems, many people who work from home choose to arrange a space to be used during business hours to delimit work from what household chores mean [16]. 

*Psychological well-being* refers to the expectations and perceptions that the individual has, taking into account their own aspirations and values. This is recognized as an element that creates in individuals a good predisposition towards the organization in order to perform at work [55]. Psychological development makes the individual inclined towards *autonomy* in performing tasks, because people of this type tend to evaluate themselves according to their own standards, not considering the collective opinion [56]. 

*Self-isolation and loneliness* can lead to greater separation from others and, over time, can negatively affect mental and physical health [57,58]. Additionally, loneliness was found to be associated with heightened danger perception and higher stress, raising the possibility that lonely people may perceive the epidemic scenario more adversely and experience higher levels of discomfort [59]. 

*Stress* refers to a complex pattern of emotional states, physiological reactions, and related thoughts, occurring in response to external demands [28] (p. 242). Stress can produce unwanted effects on health (diseases, burnout, etc.) [28] (pp. 236–237). In WFH environments, stress can increase due to the fact that many employees who work remotely must manage both the responsibilities of their professional lives and those of parenthood and other family obligations [60]. Chu AMY, Chan TWC, and So MKP [61] applied the stress mindset theory to study the relationships between three stress relievers (company support, supervisor trust, and work-life balance) on the positive and negative sides of employees’ psychological well-being, which in turn affected their job performance (productivity and non-work-related activities during working hours) when they were working from home during the COVID-19 pandemic, and shows that when employees feel happy in their WFH arrangements, their work productivity increases. To evaluate the level of stress Weinert C., Maier C., and Laumer S., showed that isolation and information undersupply of telework-characteristics are significant predictors of work overload, work-home conflict, and the role of ambiguity in the teleworking context [52] (p. 1417). In addition, the findings indicate that organizations should strengthen teleworkers’ autonomy in order to lower the impression of telework-enabled stresses, as well as the associated stress and determination to discontinue teleworking [52].

Starting from these findings, the following research questions are proposed to be answered in our research:

RQ1. What job factors are best explaining the social and technical performance in WFH?

RQ2. Which is the impact of the relevant job factors on employees’ self-assessed social and technical performance, taken as average? 

RQ3. Which is the effect of the relevant job factors for employees with different levels of assessed performance?

## 3. Data and Methods

WFH (also known as teleworking, remote working) has been defined as a system of work organization through which employees regularly fulfill their professional duties in another place than the workplace, organized by the employer, by means of information and communication technology [12]. 

### 3.1. Sampling

The survey was carried out within October–November 2021, and targeted full-time employees working from home in the past 12 months at least 20% of the total working time (one day per week). 

Prior to beginning the data collection, the questionnaire was pre-tested with four participants. The pre-testing interviews lasted on average 60 min. The participants were asked to evaluate the wording of the questions, if the response options were clear enough, and if the response options covered all the possible situations. Attention was given to how long the respondent took to answer each question, and whether they appeared to struggle to understand what the question asked. To avoid missing data, the online platform did not allow the participants to move from one question unless an answer was given. Therefore, during the pre-testing phase, special attention was given to how comfortable the participants felt while completing the questionnaire, in order to avoid the risk of providing a random answer to move on. After the pre-testing stage, the questionnaire was revised and shortened. 

The survey was scripted in NIPO software (NIPO Company, Nfield software, version Nipo Odin 5.17, Amsterdam, The Netherlands); the average length of an interview was about 22 min. The survey was conducted through the Daedalus Online access panel, the largest online panel providers in Romania. 

To obtain a sample that is representative for the population under investigation, we used quota sampling. Quotas were set for age (four age groups), gender, region, region size (four urbanization categories), and activity domain. Information regarding the demographic structure of teleworkers was obtained from the SNA Focus study coordinated by BRAT (Romanian Joint Industry Committee for Print and Internet)—a nation-wide survey carried out between June–December 2020. Survey invitations were sent gradually in the panel, but for all demographic groups simultaneously, to ensure a balanced completion of the quotas. The response rate was 19%.

The final sample consisted of 801 employees, who fit their respective target definitions. The sample profile is presented in Appendix A.

Response validations were carried out in real time during data collection and inconsistencies were removed.

Near the end of the fieldwork, some demographic quotas were relaxed. The sample of employees was then weighted to bring it to the demographic structure defined by the quotas. Weighting was completed with the SPSS RAKE module, with an 89% efficiency.

### 3.2. Response Variables (RVs) 

Job performance was measured through social performance and technical performance by adapting the battery developed by Abramis [13]. Employees were asked to self-evaluate their social and technical performance in the WFH context, on a seven-point scale, from “very poor” to “very well”. The first seven items in the battery measured technical performance, while the latter three measured social performance (Table 1). 

The descriptive statistics for social and technical performance variables can be found in Appendix B. Both variables are scores determined through confirmatory factor analysis as they are latent variables. 

The results (Figure 1) reveal that the two factors (social performance and technical performance) suggested by the theory are also identified in the structure of our data. Since there is no universally accepted criterion to judge the adequacy of the specified model, several large-scale fit indices were calculated. The main criteria used to judge model fit on the data, include the comparative fit index (CFI), proposed by Bentler [62], the Tucker-Lewis index (TLI) [63], root mean square error of approximation (RMSEA) [64] for which *p*-value has to be greater than 0.05 to conclude that the fit of the model is “close”, and the standardized root mean square residual (SRMSR) [65]. The use of the chi-square statistic is avoided as it is strongly influenced by sample size [66,67]. Hu and Bentler [68] propose as guideline values the following thresholds: RMSEA ≤ 0.06, CFI ≥ 0.95, TLI ≥ 0.95); Browne and Cudeck [69]; Jöreskog & Sörbom [65] suggested that an RMSEA value < 0.05 indicates a “close fit” and that of <0.08 suggests a reasonable fit between model and data; Byrne [70] suggests that models with SRMR values below 0.05 threshold are considered to indicate good fit, also, values up to 0.08 are acceptable [71].The values of the main criteria used to judge model fit on the data are: RMSEA = 0.061 with 90% confidence interval (0.050; 0.072), a p-close = 0.049, CFI = 0.978, TLI = 0.971, SRMR = 0.024. These values indicate that the model specified provide a satisfactory fit to the data, fit indices meet the minimum acceptability thresholds, except for the RMSEA index for which its value indicates a reasonable model fit, but the pclose is no greater than 0.05. This pclose measure consists of a one-sided test of the null hypothesis that RMSEA is less than or equal to 0.05, which is called a close model fit. Thus, having a pclose < α = 0.05 concludes that the model fit is less than a close fit (i.e., statistically significant). No further modifications were performed on the model to achieve a better fit. Theoretical consistency was considered more important than adjusting the postulated model in order to improve fit, which is determined statistically rather than theoretically.

### 3.3. Explanatory Variables (EVs) 

Based on literature, and also on the findings from the previous qualitative study, the following explanatory variables were selected [12]:Employee’s profile variables:
⇨Socio-demographic characteristics: gender, age, education level, relationship status, household size, number of children in household regardless of age; ⇨Job characteristics: professional domain, experience in WFH before the pandemic, time worked from home in a regular week (share from 40 h), time worked for current employer, work experience in the current role regardless of employer.
Organizational environment variables:
⇨*WFH engagement* was measured based on a battery developed by Schaufeli, W. B., Shimazu, A., Hakanen, J., Salanova, M., De Witte, H. [72] and refers to how enthusiastic and energetic the employees feel about their job;⇨*Occupational self-efficacy* (*effectiveness*), evaluated, based on the Self-Efficacy Scale developed by Rigotti, T., Schyns, B., Mohr, G. [27], how confident the employees are in their ability to cope with difficult tasks or problems or in their ability to successfully fulfil a task; ⇨*Computer-mediated (C-M) communication* skills evaluated, based on the C-M communication competency scale created by Spitzberg [25], how people use various online communication technologies (for example, instant messaging, email, video conferencing, chat apps, etc.) in communicating with co-workers, and if online interactions are more productive than face-to-face interactions; ⇨Motivation was evaluated based on a battery developed by Tremblay, M. A., Blanchard, S. T., Pelletier, L. G., Villeneuve, M. [30] and covers introjected, integrated and amotivation. *Introjected motivation* is a type of internal motivation, and results from feeling pressure to perform well at a job, coupled with feeling shameful when the performance is not up to par. *Integrated motivation* is about identifying with the work itself and *amotivation* refers to the lack of both internal and external motivation;⇨*Job interdependence* (received and initiated) evaluated if specific jobs are dependent on other activities in the organization, and was measured base on a battery developed by Morgeson and Humphrey [73];⇨*Situational constraints and organizational influencers* evaluated how specific constraints specific to the workplace (furniture, IT infrastructure, documentation, video-conferencing etc.) influence activity and job performance. This variable was suggested by the previous qualitative study [12];⇨*Organizational commitment* evaluated, based on a battery developed by Allen, N. J., Meyer, J. P. [35] how loyal the employees are to the company; ⇨*Interpersonal trust (trust in management and trust in co-workers)* meant in our study employees’ trust in supervisors, enforcing the importance of building work-environments, where employees are taught to rely on each other and are praised for their achievements as a team. These characteristics were separately measured, based on a battery developed by Cook, J. D. and Wall, T. D. [45];⇨*Performance reviews* identified if the employees participated in the performance reviews, and how these were organized: formally—following a clear set of criteria and pre-set objectives; semi-formally—some indicators are discussed, but the emphasis falls on what the employee does well; or informally—when ideas, opinions, subjective work perceptions, strong and weak points are discussed.
Work-life balance variables:
⇨*Family—work conflict* and *work—family* conflict evaluated the extent to which employees feel that family-related activities interfere with their jobs, and are based on the Family-Work Conflict Scales, developed by Netemeyer, R. G., Mcmurrian, R. C., Boles, J. [74];⇨*Professional isolation* assessed the extent of professional isolation experienced by employees working remotely and was measured based on a battery developed by Golden, T. D.; Veiga, J. F.; Dino, R. N. [75];⇨*Stress* was interpreted as workers’ perception of exhaustion and fatigue due to WFH and was measured based on a battery developed by Weinert, C., Maier, C., Laumer, S. [52];⇨*Job satisfaction* was measured based on a batery developed by Brayfield, A. H. and Rothe, H. F. [76], and evaluated how happy the employees felt at their job, and how much they enjoyed working in the organization;⇨*Job autonomy*, measured with a battery developed by Morgeson and Humphrey [73], evaluated the freedom of the employees in organizing their work and in the decision about the methods used or completing their tasks.


## 4. Results

In order to have a clearer view of the data, a regression model has been estimated for each dependent variable and possible predictors, for which we have carried out several post-estimation commands that can help us identify outliers. To illustrate the results, we used a leverage-versus-residual-squared plot, a graph of leverage against the (normalized) residuals squared (Figure 2). Following the analytic representation, it is evident that units 396 and 415 are extreme outliers and, therefore, have been excluded from the dataset.

Since the number of predictors is quite large, the backward stepwise selection was used in our analysis. The result (and answer to *RQ1*) is a reduced regression model consisting of explanatory variables that best explain the data, and which are significantly predictive of the response variables (Appendix C). 

In order to provide the answer for *RQ2*, we calculated the impact of the relevant job factors (explanatory variables) on employees’ self-assessed social and technical performance, taken as average (Table 2 and Table 3).

An examination of Table 2 clearly shows that from the saturated model, backward stepwise regression, has resulted in a reduced model consisting of eight independent variables that best explain the data, and which are significantly predictive of the dependent variable social performance. The variables that best explain social performance are: C-M communication skills (positive effect), trust in management (negative effect), job autonomy (positive effect), occupational self-efficacy (positive effect), family—work conflict (negative effect), engagement in WFH (positive effect), PhD studies (negative effect), and trust in colleagues (positive effect).

In terms of technical performance (Table 3), the predictors that have a significant impact are: job autonomy (positive effect), family—work conflict (negative effect), having children in the household (0–6 y.o.) (negative effect), organizational commitment (negative effect), work engagement (positive effect), stress (negative effect), household size four persons (positive effect), introjected motivation (positive effect), amotivation (negative effect), integrated motivation (negative effect), occupation self-efficacy (positive effect), trust in co-workers (positive effect), and share of time spent working from home (positive effect).

The ordinary least squares (OLS) regression technique used in our analysis in the first stage provided summary point estimates that calculate the average effect of the explanatory variables for the “average individual”. The focus on the mean alone may obscure essential features of the underlying relationship. In order to display a more complete picture of the factors affecting social and technical performance in a WFH setting, detected by backward selection, we turn to quantile regression [77]. In contrast to conventional regression models, quantile regressions can represent the entire conditional distribution of the response variable. It is a regression technique that allows one to focus on the effects that the explanatory variables have on the entire conditional distribution of the explanatory variable, namely, it takes into account that this effect can be different for employees with different levels of performance. Generally, extreme, or minimal performance cases, are of interest in their own right and, rather than dismissing these cases as “outliers”, we believe it would be worth studying them in more detail. This can be done by calculating coefficient estimates at different quantiles of the conditional distribution of performance (i.e., conditional on the explanatory variables).

In addition, the quantile regression approach avoids the restrictive assumption that the error terms are identically distributed at all points of the conditional distribution. Relaxing this assumption allows us to recognize individual heterogeneity, and to consider the possibility that the estimated slope parameters vary at different quantiles of the conditional productivity distribution. The Breusch-Pagan test is significantly different from zero (Prob > chi2 = 0.0000), for both the social and technical performance models. Also, the Shapiro-Wilk W test for normal data (Prob > z = 0.000) shows that data is not normal, which is also proved with the JB (Jarque–Bera) test. Therefore, the use of quantile regression is justified instead of OLS, as heteroscedasticity is present. A notable feature is that quantile regression does not make assumptions about the distribution of the target variable and resists the effects of outlying observations.

In order to carry out the analysis, and provide the answer to *RQ3*, we first point out that two types of significances are important for quantile regression coefficients: coefficients significantly different from zero; and coefficients significantly different from OLS model, which show different effects along the distribution. In this study, quantile regression distributes the respondents’ performance into 10th, 25th, 50th, 75th, and 90th quantiles. The 10th quantile refers to the 10% of the respondents who are least productive working from home, while 90th quantile refers to the 10% of the most productive respondents. 

### 4.1. The Social Performance Model

The quantile regression model for social performance is based on Equation (1):(1)$$Q_{q}\left(SP\right)=\sum_{i=1}^{8}\left(\propto_{q}+\beta_{q,i}\;\times \;SP\right)+\varepsilon$$
where:
*Q*—quantile for the explanatory variable (Iv)*q* = 0.1, 0.25, 0.5, 0.75 or 0.9*SP*—Social Performance Explanatory Variable (please see Table 4)*i* = 1–8—Number of the explanatory variables

The analysis results are presented in Table 4 and Figure 3 and illustrate the distribution of the OLS and QR (Quantile Regression) coefficients for different explanatory variables. We should remember that OLS coefficients are determined by changes in conditional mean, whereas QR coefficients are determined by changes on given conditional quantile. In each panel, the horizontal axis displays the different quantiles, and the vertical axis displays the effect of each explanatory variable, holding other covariates fixed. OLS coefficients are represented by a solid line parallel to the horizontal axis, respectively, and their confidence intervals by dashed lines. Grey areas in each panel correspond to the confidence intervals of the conditional quantiles.

### 4.2. The Technical Performance Model

In terms of technical performance, the model for the selected quantiles is shown in Equation (2), while the results are presented in Table 5 and Figure 4:(2)$$Q_{q}\left(TP\right)=\sum_{i=1}^{14}\left(\propto_{q}+\beta_{q,i}\;\times \;TP\right)+\varepsilon$$
where:
*Q*—quantile for the explanatory variable (Iv)*q* = 0.1, 0.25, 0.5, 0.75 or 0.9*TP*—Technical Performance Explanatory Variable*i* = 1–14—Number of the explanatory variable

## 5. Discussions and Recommendations for Organizations

The influence of the WFH factors on social and technical performance is summarized in Figure 5.

### 5.1. WFH Factors Predicting Both Social and Technical Performance

*Engagement* is a strong positive predictor for both social and technical performance, confirming previous studies that showed that the employees who feel more engaged and enthusiastic when working remotely are more productive [24] (p. 9). 

Our research brings more insights into these findings, showing that the effect of WFH engagement is stronger for employees with lower performance levels and is decreasing for respondents with higher performance, belonging to the 75th and 90th quantile levels. Fostering a work environment where employees feel that they thrive (through implementing training programs that inspire and embolden them, for instance), is a strategy that is worth keeping in mind for organizations. Some employees may simply feel more energy, strong involvement, and complete focus [22,23], when working from home, which in turn boosts their productivity—this could be due to them finding joy in working in a familiar environment or, conversely, dreading the commute to the office.

*Effectiveness* (*Occupational self-efficacy*) is a strong positive predictor for both social and technical performance across all quantiles, this result being consistent with previous studies performed in WFO environments [27] (p. 250). Building confidence and empowering employees through providing positive feedback and praising their work when they perform well, coupled with offering constructive criticism rather than critique when they stumble, is a strategy worth adopting by companies, especially in the context of a remote work setup. Work effectiveness coefficients are significantly different from OLS regression at the 50th and 90th quantile levels, but also significantly different from 0, for both social and technical performance. If we look at Figure 3 and Figure 4, we can observe a siderite effect along the social and technical performance distribution. We also observe that at a one unit increase in effectiveness, technical performance increases by 0.5347 units and social performance increases by 0.4115 for those with medium performance (at the 50% quantile). It is evident that the effect of effectiveness is more significant for those with medium scores, while for those with high scores the effect diminishes (higher quantiles), and this finding represents another contribution to the relevant literature.

*Job autonomy* has a positive effect on both social and technical performance, with the highest levels at employees with the lowest performance scores and with the lowest level from the employees from the 0.75 quantile. Autonomy is important, because it makes employees exercise more control over their work, leading to motivation, better performance, commitment, and satisfaction [73,78,79]. 

*Family-work conflict* is a negative predictor for both social and technical performance with the highest levels at employees with the lowest performance scores. Combining work and family demands into one physical space can lead to several negative outcomes, such as lower job satisfaction and performance [53,80]. 

*Trust in co-workers* is positively related to performance, confirming the previous findings from the literature [42,43]. This factor strongly influences employees with low levels of social and technical performance, after which there is a diminishing effect for those with a high-performance level. It is obvious that the employees situated in the lower-level categories are the ones who rely on their workmates when difficulties arise, being confident in the skills of their colleagues.

*C-M communication skills* show a fairly uniform distribution, according to the quantile regression results for both social and technical performance, with the highest level at employees with the lowest performance scores.

### 5.2. WFH Factors Predicting Social Performance

We found that *trust in management* generates a negative effect on social performance. This relationship is explained by a compensation effect—in WFH environments where the management is weak; employees are forced to rely on each other more and communicate more closely to get things done. This in turn strengthens the relationships within teams and also urges employees to find ways of efficiently dealing with any tensions that might develop themselves (without the involvement of management). On the other hand, trust in management has no effect on technical performance, confirming the previous studies, which focused mainly on technical aspects of performance, that showed that *trust in management* generates positive or no effect on job performance [44,48]. 

The level of *PhD education* is negatively associated with respondents with low social performance (i.e., Q 0.1). The estimated performance of individuals with doctoral degrees will be reduced by 0.5431 over those without doctoral degrees. In addition, at the 0.90 quantile, there is a substantially different quantile regression coefficient from OLS coefficient, which indicates that at the level of respondents with very high social performance scores, the effect of doctoral education is diminished.

### 5.3. WFH Factors Predicting Technical Performance

Surprisingly, *organizational commitment* has a significant negative effect on reported technical performance, with the highest levels on employees with the lowest performance scores, in opposition to previous studies [36,37,38]. It is worth mentioning here that the previous studies were performed in WFO environments and that performance was evaluated globally, with no delimitation between social and technical performance. The negative effect of *organizational commitment* on reported technical performance could be attributed on one hand to the employees’ own standards and expectations about their performance, and the desire to impress management. On the other hand, employees who feel more strongly committed to their organizations may thrive at their jobs when working from the office, rather than from home. 

Both QR and OLS results show that *stress* negatively impacts employees’ technical performance when working from home, being on top of the important predictors, like previous studies demonstrated [52,61]. According to the QR results, stress is fairly uniform distributed, with the highest level for the 0.10 quantiles, which indicates that at the level of respondents with low technical performance scores, the effect of stress is more significant. Lower levels of stress are indicted for employees with median and highest level of technical performance. Paying close attention to employees’ stress levels, monitoring their workload and alleviating pressure when they cannot cope or are overwhelmed with tasks is paramount to building a healthy work environment, and should be a priority for any people management strategy.

When it comes to motivation, *introjected motivation* has a positive effect on technical performance, with the highest levels being among employees with the lowest performance scores. This type of employee thrives when their efforts are recognized and praised, however, they might become too hard on themselves when they slip up. On the flipside, *integrated motivation* and *amotivation* have a significant negative effect on technical performance, being more prominent in the employees with high performance levels.

Employees who worked from home a higher *share of their work time* report higher levels of technical performance, which indicates that the more employees work from home, the more likely they are to settle into the new work setup and develop routines and strategies that enables them to work productively, as opposed to those employees who only occasionally make their homes into their office.

Having *young children in the household* (6 y.o. or younger) negatively impacts technical performance—this could be a direct effect of the challenges of splitting the workday between job and family responsibilities, and the pressures of caring for toddlers while also fulfilling job tasks.

## 6. Conclusions, Limitations and Future Research

We investigated several faces of the WFH system showing that job performance is a complex concept that is influenced by individual, organizational environment, and work-life balance factors. Job performance is usually analyzed in literature in general terms, with no distinction between social and technical performance [1,10,12].

On the other hand, the models and methods used in previous studies provided summary point estimates that calculated the average effect of the explanatory variables for the “average individual” [10,15,22].

The main shortcoming of these approaches is that the mean alone may obscure important features of the underlying relationship, and the effects of the explanatory variables can be different for employees with different levels of performance. Generally, extreme, or minimal, performance cases are of interest in their own right and, rather than dismissing these cases as “outliers”, we believe it would be worth studying them in more detail. 

This study brings contributions in the field of remote working, a research field that receive a huge attention after the COVID-19 pandemic [8,9,21,24]. Our findings are bringing significant insights into the factors that influence the technical and social self-assessed performance of the employees working in remote systems. These factors are influencing differently the employees at average performance level (as resulted from OLS analysis) and the employees self-assessed with higher or lower performance (as resulted from QR analysis).

Several limitations need to be acknowledged. First, this study relies on cross-sectional survey data with self-reported measures. Therefore, the data do not permit analysis behavior over a period of time. The sample consisted of Romanian employees, mostly engaged in the so-called “knowledge work”. In addition, the respondents had to have the available means to participate in the study (Internet connection and time). Therefore, the study results may suffer in case of other types of occupation or other socio-economic contexts, differently affected by the pandemic. At the same time, the type of remote work investigated in the study was imposed by the pandemic context in many cases. The data were collected at the end of the pandemic, so people had time to adjust and find ways to meet challenges associated with teleworking. Therefore, it is not guaranteed that the situation will be the same in the long run, in a non-pandemic environment, or in other types of “forced” remote work systems. 

The results are of particular interest for researchers and for the managers of organizations. Once these factors and influences are understood, the managers could find better tools to adapt to these new challenges, and create an environment enjoyed by employees, where they could perform better.

Literature provides plenty of studies on job performance and its predictors, the majority of them focusing on traditional WFO environments. Future studies must explore if the findings can also be applied in the new WFH scenarios. 

For future research it will be interesting to extend such analyses in other countries. It will be also interesting to see how employees belonging to different activity domains are influenced by job factors, and also to identify different clusters of teleworking behavior patterns. Another interesting research direction would be to change the perspective of assessment, from employees to managers.

## Figures and Tables

**Figure 1 ijerph-19-10935-f001:**
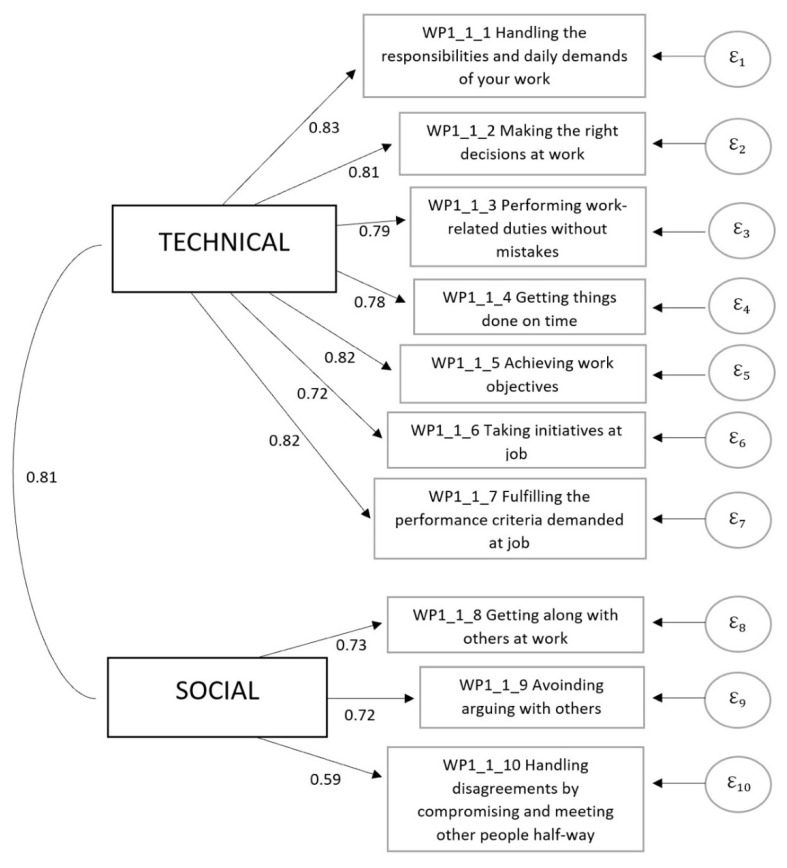
Self-assessed social and technical performance.

**Figure 2 ijerph-19-10935-f002:**
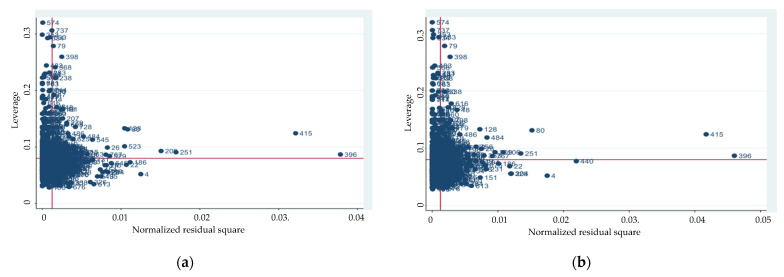
Leverage-versus-residual-squared plot: (**a**) for social performance; (**b**) for technical performance.

**Figure 3 ijerph-19-10935-f003:**
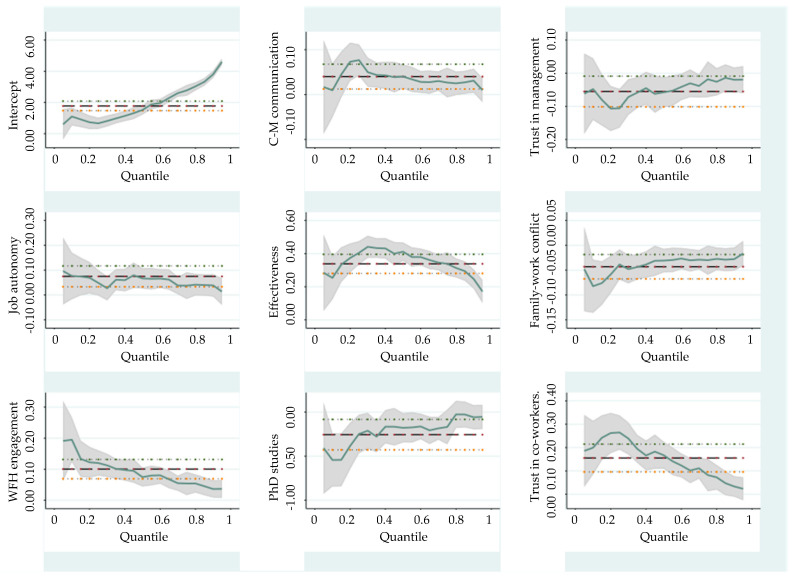
OLS and QR estimates for social performance model.

**Figure 4 ijerph-19-10935-f004:**
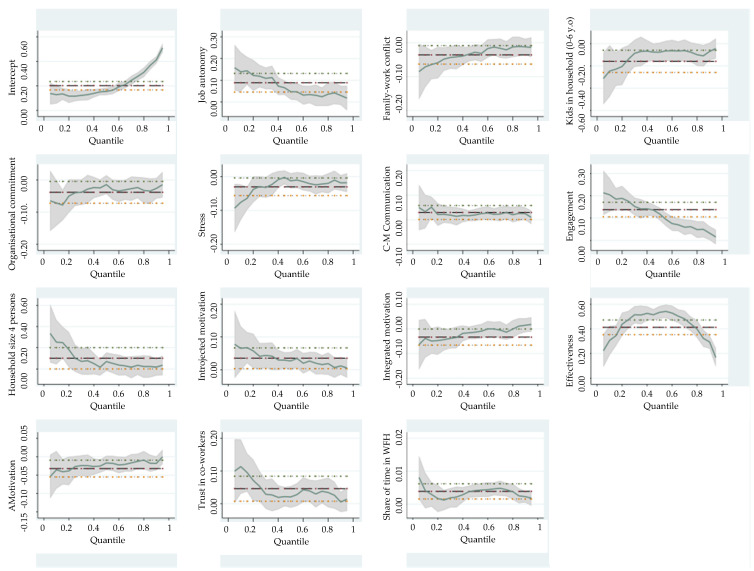
OLS and QR estimates for technical performance model.

**Figure 5 ijerph-19-10935-f005:**
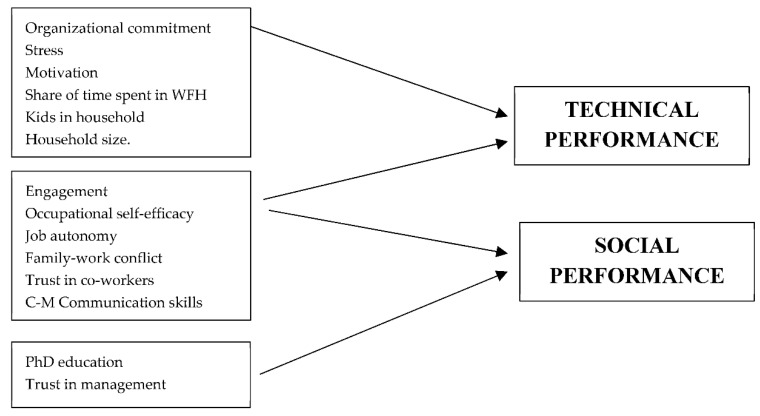
The influence of the WFH factors on social and technical performance.

**Table 1 ijerph-19-10935-t001:** Response variables (technical and social performance).

Response Variable	Code
**Technical performance**	
Handling the responsibilities and daily demands of your work	WP1_1_1
Making the right decisions at work	WP1_1_2
Performing work-related duties without mistakes	WP1_1_3
Getting things done on time	WP1_1_4
Achieving work objectives	WP1_1_5
Taking initiatives at job	WP1_1_6
Fulfilling the performance criteria demanded at job	WP1_1_7
**Social performance**	
Getting along with others at work	WP1_1_8
Avoiding arguing with others	WP1_1_9
Handling disagreements by compromising and meeting other people half-way	WP1_1_10

**Table 2 ijerph-19-10935-t002:** Multiple linear regression analysis results related to social performance—backward selection.

Category	Explanatory Variables	Coef.	Std. Err.	t	P > t	[95% Conf. Interval]	
Individual employee profile variables	PhD studies	−0.2555	0.0877	−2.91	0.004	−0.4278	−0.0832
Organizational environment variables	WFH engagement	0.1003	0.0159	6.31	0.000	0.0691	0.1315
	Occupational Effectiveness	0.3380	0.0293	11.54	0.000	0.2805	0.3955
	C−M communication skills	0.0400	0.0140	2.84	0.005	0.0123	0.0676
	Trust in management	−0.0550	0.0235	−2.34	0.020	−0.1012	−0.0087
	Trust in co−workers	0.1555	0.0303	5.12	0.000	0.0957	0.2147
Work−life balance variables	Familywork conflict	−0.043	0.0124	−3.45	0.001	−0.0674	−0.0185
	Job autonomy	0.0752	0.0214	3.50	0.000	0.0330	0.1173
	Intercept	1.7782	0.1552	11.45	0.000	1.4734	2.0831
	R^2^ = 0.4903 F(8,790) = 94.99 *p* = 0.000						

**Table 3 ijerph-19-10935-t003:** Multiple linear regression analysis results related to technical performance—backward selection.

Category	Explanatory Variables	Coef.	Std. Err.	t	P > t	[95% Conf. Interval]	
Individual employee profile variables	Household size (4 persons)	0.1005	0.0510	1.97	0.049	0.0003	0.2008
	Kids in household (0–6 y.o.)	−0.1586	0.0509	−3.12	0.002	−2.586	−0.0587
	Share of time spent working from home	0.0019	0.0005	3.30	0.001	0.0007	0.0030
Organizational environment variables	WFH engagement	0.1381	0.0166	8.31	0.000	0.1055	0.1707
	Occupational Effectiveness	0.4127	0.0302	13.63	0.000	0.3532	0.4719
	C-M communication skills	0.0293	0.0144	2.03	0.043	0.0098	0.0576
	Introjected regulation	0.0349	0.0160	2.18	0.030	0.0034	0.0664
	Integrated motivation	−0.0334	0.0166	−2.01	0.045	−0.0661	−0.0007
	Amotivation	−0.0322	0.0115	−2.79	0.005	−0.0548	−0.0095
	Organizational commitment	−0.0382	0.0173	−2.21	0.028	−0.0723	−0.0042
	Trust in co-workers	0.0458	0.0194	2.36	0.019	0.0076	0.0840
Work−life balance variables	Family—work conflict	−0.0358	0.0139	−2.57	0.010	−0.0631	−0.0084
	Job autonomy	0.0886	0.0217	4.08	0.000	0.04594	0.1312
	Stress	−0.0301	0.0132	−2.27	0.024	−0.0561	−0.0040
	Intercept	2.0193	0.1748	11.55	0.000	1.6760	2.3626
	R^2^ = 0.5787 F(15,783) = 71.71 *p* = 0.000						

**Table 4 ijerph-19-10935-t004:** Analysis results of least square regression model and quantile regression model for social performance.

Category	Explanatory Variables	OLS Regression	0.10 Quantile	0.25 Quantile	0.50 Quantile	0.75 Quantile	0.90 Quantile
Individual employee profile variables	PhD studies	−0.2555 ***	−0.5431 ***	−0.2485	−0.1784 *	−0.1684	−0.0585
Organizational environment variables	WFH engagement	0.1003 ***	0.1950 ***^,+^	0.1199 ***	0.0745 ***	0.0540 ***^,+^	0.0362 ***^,+^
	Occupational Effectiveness	0.3380 ***	0.2537 ***	0.4048 ***	0.4115 ***	0.3383 ***	0.2470 ***^,+^
	C-M communication skills	0.0400 ***	0.0096	0.0767 ***	0.0400 ***	0.0267 ***	0.0309 ***
	Trust in management	−0.0550 **	−0.0479	−0.1057 ***^,+^	−0.0576 **	−0.0181	−0.0197
	Trust in co-workers	0.15552 ***	0.19966 ***	0.2646 ***^,+^	0.1674 ***	0.0821 **^,+^	0.03328
Work-life balance variables	Family—work conflict	−0.043 ***	−0.0822 ***	−0.0385 **	−0.0307 ***	−0.0298 **	−0.0276 ***
	Job autonomy	0.0752 ***	0.0768	0.0486 *	0.0667 ***	0.0378	0.0387 **
	Intercept	1.7782 ***	1.0982 ***^,+^	0.6629 ***^,+^	1.5201 ***	2.7714 ***^,+^	3.8044 ***^,+^
	Model Summary						
	Pseudo R^2^	-	0.3275	0.3403	0.3116	0.2674	0.1524

Note: * is significant at the 10% significance level; ** is significant at the 5% significance level; *** is significant at the 1% significance level; ^+^ Significantly different quantile regression coefficients from OLS coefficients at the 5% significance level, when the OLS coefficient is outside of the quantile regression coefficient confidence interval.

**Table 5 ijerph-19-10935-t005:** Analysis results of least square regression model and quantile regression model for technical performance.

Category		OLS Regression	0.10 Quantile	0.25 Quantile	0.50 Quantile	0.75 Quantile	0.90 Quantile
Individual employee profile variables	Household size-4 persons	0.1005 **	0.2522 ***	0.1003 *	0.0687	0.0331	0.0174
	Kids in household (0–6 y.o.)	−0.1586 ***	−0.2455 **	−0.1399 **	−0.0766	−0.0762 **^,+^	−0.0701
	Share of time spent working from home	0.0019 ***	0.0020	0.0006	0.0020 ***	0.0021 ***	0.0011 **
Organizational environment variables	WFH engagement	0.1381 ***	0.2045 ***	0.1756 ***	0.1191 ***	0.0614 ***^,+^	0.0327 *
	Occupational Effectiveness	0.4127 ***	0.3062 ***	0.4750 ***	0.5347 ***^,+^	0.4403 ***	0.2935 ***^,+^
	C-M communication skills	0.0293 **	0.0328	0.0216	0.0191 *	0.0292 *	0.0247 ***
	Introjected motivation	0.0349 **	0.0650 **	0.0408 **	0.0265 *	0.0153	0.0112
	Integrated motivation	−0.0334 **	−0.0386 *	−0.0438 **	−0.0133	−0.0134	0.0147
	Amotivation	−0.0322 ***	−0.0356 *	−0.0267	−0.0170 **	−0.0127	−0.0182 **
	Organizational commitment	−0.0382 **	−0.0715 **	−0.0397 *	−0.0142	−0.0234	−0.0261
	Trust in co-workers	0.0458 **	0.1134 ***	0.0531	0.0206	0.0380 ***	0.0052
Work-life balance variables	Family—work conflict	−0.0358 **	−0.0713 ***	−0.0483 ***	−0.0279 **	−0.0209 *	−0.0121
	Job autonomy	0.0886 ***	0.1389 **	0.1189 ***	0.0475	0.0242	0.0302
	Stress	−0.0301 **	−0.0758 **	−0.0293	−0.0123	−0.021 *	−0.0184
	Intercept	2.0193 ***	1.2905 **	1.1662 ***^,+^	1.5524 ***	2.7687 ***^,+^	4.1130 ***^,+^
	Model Summary						
	Pseudo R^2^	−	0.4551	0.4218	0.3864	0.2951	0.1610

Note: * is significant at the 10% significance level; ** is significant at the 5% significance level; *** is significant at the 1% significance level; ^+^ Significantly different quantile regression coefficients from OLS coefficients at the 5% significance level, when the OLS coefficient is outside of the quantile regression coefficient confidence interval.

## Data Availability

Not applicable.

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
