# Peer review of "Work from Home during the COVID-19 Pandemic—The Impact on Employees’ Self-Assessed Job Performance"

_ijerph, 2022, doi:10.3390/ijerph191710935_

Round 1
Reviewer 1 Report
Overall comment on this research article “Work from Home during the COVID-19 Pandemic: The Impact on Employees’ Self-Assessed Job Performance” is very positive. Comprehensive data is collected (801 respondents) and analyzed in a very systematic way to find the impact.
The author(s) presented a very impressive literature review to develop a context for Work from home and discussed it in detail with very relevant references.
Author Response
Dear Reviewer,
We are grateful for your comments.
The authors
Reviewer 2 Report
It is a study whose topic is current and important, both theoretical and practical.
It shows an important theoretical support with previous studies.
The method is consistent and consistent with what was wanted to know, with a combination of statistical analysis.
The results show the relevant findings of the above, in addition, results are discussed, in which it may be possible to expand with further studies to be discussed.
It is a study whose topic is current and important, both theoretical and practical.
It shows an important theoretical support with previous studies.
The method is consistent and consistent with what was wanted to know, with a combination of statistical analysis.
The results show the relevant findings of the above, in addition, results are discussed, in which it may be possible to expand with further studies to be discussed.
The conclusions are pertinent and derived from results.
Carry out a discussion of the results in relation to previous studies or findings derived from other studies, in order to analyze similarities and differences, in such a way that the theoretical-empirical support of the research is strengthened.
Author Response
Dear Reviewer,
We are grateful for the comments that help us improve the paper. We made the necessary changes and we consider that the new version of the paper is meeting the requirements to be published. All suggestions have been completed. The explanations are provided as follows:
-Section 2 (References) was updated with new references
-Section 5 (Discussions) was completely re-written, in this version we discuss the results in relation to previous studies.
-Section 6 (Conclusion) was also improved
The authors

Reviewer 3 Report
Thank you for the opportunity to review this interesting and well-written manuscript. Authors report the results of a survey of 80 working from home Romanian employees, aimed at identifying the factors associated with remote working which can have an impact on employees’ job performance (both social and technical).
The study is of relevance due to the current debates about the pros and cons of working from home, and whether it should become the “new normal” even after the Covid-19 pandemic. Although the study has merits, I have a few methodological/presentation points, which I encourage the authors to address.
1. Please add also the response rate to the “Sampling” section;
2. It would be useful to have also some basic statistics about the explanatory variables (Cronbach’s alpha, Mean, and Standard Deviation, at least). I also believe that many of these variables are inter-correlated, I’d suggest reporting also the correlations among all the measures;
3. Some explanatory measures (i.e., Job interdependence, Situational constraints, Performance reviews, Professional isolation, and Job satisfaction) are not included in Appendix C. Moreover, some items of the Motivation scale have a strikethrough formatting, please check.
4. Please provide a more detailed explanation about how Figure 2 graphs answer the RQ1;
5. Backwards stepwise regression is controversial at best, and it is usually suggested not to use it. The problem, as stated by Smith (2018), is that “some real explanatory variables that have causal effects on the dependent variable may happen to not be statistically significant, while nuisance variables may be coincidentally significant.”. I’m not totally against this regression model but, due to the strong and widespread criticism against it, the authors should better justify the rationale behind their choice. Perhaps finding strong correlations among the explanatory variables (see point 2 above) could help;
6. The results displayed in Tables 2 and 3 should provide the answer for RQ2, nevertheless, they are not commented on at all;
7. On page 17, the authors write “Family-work conflict is a negative predictor for both social and technical performance, confirming the hypothesis that some employees would benefit from having a stronger separation between the two areas of their lives.”. Is it possible to identify who are these employees? I believe that workers with young kids in the household (another relevant explanatory variable reported on the same page) could be a good candidate. I wonder whether the authors could dig more into this with their data, especially because the relationship between family-work conflict and job performance is fundamental for understanding the sustainability of working from home also after the Covid-19 pandemic (in other words, if the family-work conflict is mainly due to the presence of young kids who can’t go to school/nursery due to the pandemic, than we should expect this effect to be way lower after the pandemic when parents can work freely without having to care for toddlers at the same time).
8. On page 18, no limitations are reported in the “Conclusions, limitations and future research” section”. At least, this is a cross-sectional study with self-reported measures, with all the well-known limitations of this design.
Other minor points:
- On page 4 (Family-work and Work-family conflicts), the authors write “When they fail to distinguish between the two roles, the risk of work-home conflicts is inevitable.”. This is true, but the work-home conflict happens also when the requests coming from both worlds (work and family) are not compatible;
- CFA is a well-known method, so the reporting of fit indices (page 7) can be simplified. Moreover, RMSEA is usually reported along with its 90% Confidence Interval, not with its p-value.
- On page 18, the Data Availability Statement is missing.
Author Response
Dear Reviewer,
We are grateful for the comments that help us improve the paper. We made the necessary changes and we consider that the new version of the paper is meeting the requirements to be published. All suggestions have been completed. The explanations are provided in the attached file
The authors

Reviewer 4 Report
Thank you for the opportunity to review this paper. The motivation for the study is clear. The study is timely, and the results are interesting. The methods are robust.
My main concerns are in the discussion and conclusions. In the discussion, the interpretation of results is accurate. Though, how do results compare to the current literature and knowledge on the topic? The conclusion section is very broad. I do not see any limitations mentioned in the last section either. Lastly, the manuscript has several minor grammatical errors throughout.
Please see my more detailed comments below. I hope that they will be helpful during the revision process.
Introduction:
-The introduction is well-written and provides a strong basis for the study.
-The literature review is in-depth. While the reader may appreciate the level of detail, it is a bit long. I suggest condensing and further synthesizing the literature to shorten the section.
-Is there any more academic or grey literature or reports on the impact of remote work specific to Romania? It would be helpful to include this to get a better idea if cultural or regional factors may impact results.
Methods:
-Methods are well described and robust. I appreciated the detailed explanations of the variable definitions and basis for inclusion.
-Appendix A: It's important to report the profile of part-time vs. full-time workers or the average number of hours worked per week. Perceptions may differ based on the number of hours worked per week. While I recognize that modeling accounts for this variable, it would be helpful to see it descriptively as well.
-Table 1: It is labeled as explanatory variables, though I believe that it is response variables.
-Figure 1 is a bit technical looking because of the use of variable codes as opposed to labels/names. I suggest including labels instead of codes.
Results:
-Results are very interesting and will contribute significantly to the current body of literature.
-Lines 386-387: Was there no missing data? Did every subject in the sample respond to every question? If not, please clarify the definition of “missing data.”
-General note for the result tables: I suggest adding captions indicating the control variables for modeling. It will be easier on the reader than finding it in the methods.
Discussion:
-While the authors did an excellent job interpreting the results and suggesting explanations for the results, it would be much improved if results were also placed in the context of the current literature on the topic before and after the pandemic. I do not see any references to literature in the discussion.
- As it stands, the research is interesting, but the “so what?” is less clear. What do the results indicate for policy? More organizational (or even national-level) policy discussion would greatly improve the discussion. For example, the results could be presented in the context of policy options for organizations.
Conclusions, limitations, and future research:
-The conclusions are a bit broad. Yes, they indeed bring insights to the field of remote work. But what exactly is indicated? I suggest adding a couple sentences with major takeaways to greatly improve this section.
-The areas of future research are identified though also broad.
-I don’t see any limitations listed here.
Author Response
We are grateful for the comments that help us improve the paper. We made the necessary changes and we consider that the new version of the paper is meeting the requirements to be published. All suggestions have been completed. The explanations are provided in the attached file.
The authors

Round 2
Reviewer 4 Report
Thank you for the responsiveness to the suggested revisions and comments. I am satisfied with the revised manuscript.
Author Response
Dear reviewer,
Thank you for your comments, our paper looks better now.
The authors